# Fluorescence Lifetime Endoscopy with a Nanosecond Time-Gated CAPS Camera with IRF-Free Deep Learning Method

**DOI:** 10.3390/s25020450

**Published:** 2025-01-14

**Authors:** Pooria Iranian, Thomas Lapauw, Thomas Van den Dries, Sevada Sahakian, Joris Wuts, Valéry Ann Jacobs, Jef Vandemeulebroucke, Maarten Kuijk, Hans Ingelberts

**Affiliations:** 1Department of Electronics and Informatics (ETRO), Vrije Universiteit Brussel, 1050 Brussels, Belgium; pooria.iranian@vub.be (P.I.); thomas.apauw@vub.be (T.L.); thomas.van.den.dries@vub.be (T.V.d.D.); sevada.sahakian@vub.be (S.S.); joris.wuts@vub.be (J.W.); jefvdmb@etrovub.be (J.V.); mkuijk@etrovub.be (M.K.); 2MOBI Rresearch Center, Vrije Universiteit Brussel, 1050 Brussels, Belgium; valery.ann.jocobs@vub.be

**Keywords:** endoscopy, fluorescence imaging, fluorescence lifetime imaging, gated camera, CAPS, convolutional neural networks

## Abstract

Fluorescence imaging has been widely used in fields like (pre)clinical imaging and other domains. With advancements in imaging technology and new fluorescent labels, fluorescence lifetime imaging is gradually gaining recognition. Our research department is developing the *tau*CAM^TM^, based on the Current-Assisted Photonic Sampler, to achieve real-time fluorescence lifetime imaging in the NIR (700–900 nm) region. Incorporating fluorescence lifetime into endoscopy could further improve the differentiation of malignant and benign cells based on their distinct lifetimes. In this work, the capabilities of an endoscopic lifetime imaging system are demonstrated using a rigid endoscope involving various phantoms and an IRF-free deep learning-based method with only 6-time points. The results show that this application’s fluorescence lifetime image has better lifetime uniformity and precision with 6-time points than the conventional methods.

## 1. Introduction

Optical imaging has enormous potential for the non-invasive detection and characterization of tissues utilizing non-ionizing radiation in the visible (VIS) to near-infrared (NIR) spectrum (400–1100 nm). This approach offers real-time analysis, lower cost, and nonionized compared to radiological imaging such as MRI and nuclear imaging. Despite the potential of optical imaging modalities, relatively few have reached routine use in healthcare. One of these techniques is fluorescence imaging. Fluorescence imaging has proven beneficial in different applications, such as fluorescence-guided surgery targeted to identify abnormal tissues [1]. Its high sensitivity and capability for multiplexed imaging have made fluorescence imaging an essential tool also within biomedical research, particularly in cancer studies, where it aids in understanding tumor microenvironments and mechanisms of disease progression. Fluorescence imaging provides real-time imaging capabilities, which is a significant advantage over other imaging modalities that may require postprocessing and reconstruction of images [2,3]. However, this technique relies on the intensity of the fluorophores, which correlates with the concentration of fluorophores from labeled specimens. This can lead to complicated data interpretation since the intensity can be further influenced by the local environment of fluorophores, the presence of other fluorescence materials, and by photobleaching.

In contrast, fluorescence lifetime (FLT) imaging offers the potential of enhanced contrast over conventional fluorescence intensity imaging, providing better insights that are less affected by signal intensity, fluorophore concentration, and sensitivity to chemical changes [4,5,6]. FLT is defined as the rate at which fluorescence light decays over time [7]. and there is a growing interest in leveraging this parameter for imaging to attain more profound insights into tissue characteristics and to increase contrast [8].

Fluorescence dyes can be conjugated with antibodies to create contrast agents that target specific disease markers. These contrast agents, after injection in a patient, bind to specific proteins or receptors that are (over) expressed in tumor cells. In turn, it allows the visualization of the tumor cells with the emitted fluorescence when exposed to an excitation source. Since the lifetime can show a dependency on the environment [9,10,11,12], changes in lifetime can be exploited to create additional contrast between malignant and benign tissue. Fluorescence imaging in near-infrared (NIR) exhibits a variety of (pre-) clinical applications due to high tissue penetration (millimeters to centimeters) [6], high sensitivity for real-time imaging, low cost [13,14], reduced scattering, absorption, and autofluorescence [15,16].

Laparoscopic surgeries, aided by endoscopes, have lower invasiveness compared to traditional surgeries. However, advancements in high-resolution endoscope technology, together with the increased contrast provided by fluorescence, should enable surgeons to better identify abnormal tissues during laparoscopic surgeries improving patient outcomes [17,18]. Compared to intensity imaging, FLT can provide additional contrast while mitigating some of the drawbacks, applying it to endoscopy is a logical next step.

Most NIR in vivo fluorescence imaging is performed with the support of exogenous contrast agents. However, the only ones that are currently FDA approved are indocyanine green (ICG) [19], OTL38 [20], and Lumisight [21]. In [22,23,24,25], it is reported that ICG can be used for endoscopy of sentinel lymph tumors and interoperative of certain solid tumors after intravenous injection. Burggraaf et al. [26] have compiled a great overview of NIR fluorescence endoscopy using ICG to diagnose cancer and other diseases in clinical studies.

FLT techniques can be classified based on different parameters. These classifications include frequency-domain and time-domain techniques, microscopic or macroscopic regimes, point-scanning or wide-field imaging techniques, and photon counting or analog techniques. Moreover, the nature of the technique may determine whether the signal needs to be acquired in a few time gates or in a large number of time channels and whether this acquisition occurs simultaneously or sequentially.

FLT-based endoscopy, especially in the context of surgery, requires a wide-field real-time imaging approach. As such, time-correlated single photon counting (TCSPC) [27], cannot be used due to the time required to acquire a full image. Time-gated cameras are better suited to these applications since they acquire a signal for all pixels simultaneously. However, one still needs a high-speed and NIR sensitive sensor that allows for efficient acquisition of frames. Depending on the lifetime determination method, only a few time-resolved images are sufficient, whereas other methods, like curve fitting, may require a larger number spaced over the full fluorescent decay. Intensified CCD cameras are traditionally used for wide-field FLT imaging; however, they have significant drawbacks, such as limited repetition rate, and low quantum efficiency (particularly in the NIR region) leading to significant acquisition times.

Real-time wide-field fluorescence imaging is achievable only through processing methods that can determine the lifetime of the pixels in real-time, as well as fast acquisition of the required time-resolved datapoints, preferably with a limited number of these as each acquisition adds additional latency.

This work presents NIR wide-field fluorescence lifetime endoscopy imaging with the latest generation of the state-of-the-art nanosecond time-gated *tau*CAM based on a CMOS CAPS sensor. This sensor has good NIR quantum efficiency and allows for high repetition rates, yielding reduced acquisition times. This camera is combined with a telescopic rigid NIR endoscope, and the time-domain measurements is processed using a new convolutional neural network (CNN) architecture, designed to operate only 6-time points. The CNN was trained with a fully synthetic Instrument Response Function (IRF) and the results of phantom measurements were validated against curve fitting.

## 2. *tau*CAM^TM^

The camera used in this work is the *tau*CAM, based on the latest generation of CAPS sensor with 128 × 128 pixels. The CAPS sensor is a high-speed, time-gated sensor utilizing CMOS technology, which achieves rapid in-pixel gating through its innovative topology and current-assistance mechanism. It attains a quantum efficiency of 46% in the NIR spectrum (780 nm). This sensor can gate durations as short as 1 nanosecond, with repetition rates ranging from 10 MHz to 100 MHz. More details about the *tau*CAM can be found in [28].

## 3. Experimental Setup

The system was built with a telescopic rigid NIR/ICG endoscope (26003ARA, 0° enlarged view, diameter 10 mm, length 31 cm, HOPKINS Rubina, Karl Storz, Tuttlingen, Germany), a VIS-NIR camera lens (16 mm/F1.6, 67714, Edmund optics, Barrington, IL, USA), and a single-band bandpass filter (FF01-819/44-25, Semrock). The excitation light came from a single-frequency Katana HP 35 ps pulsed laser at 775 nm (NKT Photonics, Denmark). The excitation pulse was transmitted via a fiber patch cable (numerical aperture = 0.39, core diameter 400 μm, Thorlabs, Newton, MA, USA) via a 3D printed SMA adapter to the endoscope. This pulse was then transmitted through the fiber bundle (the input port of the endoscope for the illumination consists of the fiber bundle) in the annular configuration from the endoscope distal end onto the imaging field. The emitted fluorescence signal was transmitted through the imaging part of the endoscope located in the center of the rigid tube, surrounded by the illumination fiber bundle. Figure 1 shows the FLT endoscopy setup. The used optical elements were chosen with NIR coatings to minimize losses and avoid reflection in the optical system as the sensitivity of the endoscope depends on the efficiency of light transmission from the distal end, through the endoscope, via the eyepiece to the camera.

The Hopkins Rubina endoscope is optimized for a spectral range from 400 to 900 nm; however, the built-in fluorescence excitation filter blocks the wavelength range from 670 nm to 815 nm, enabling detection of NIR fluorescence only outside that band. Thus, recording the IRF with this built-in filter is impossible for this endoscope system. A pulsed laser excitation source with a maximum average power of 1 W was used for this time-resolved application. The lower excitation power will result in a lower fluorescence emission intensity, reducing system sensitivity. Additionally, the laser’s output has a Gaussian shape and is uncollimated; without measures, this would cause non-uniform illumination of the input facet of the endoscope, resulting in inefficient light coupling. To cope with this issue, a beam collimator collates the light and changes the Gaussian shape to an almost uniform pattern, illuminating the input facet of the illumination port. Figure 2a shows the intensity of the illumination over the field of view (FoV) seen with the *tau*CAM at a working distance (WD) of 3 cm. Based on the annular configuration of the fiber bundle at the distal end of the endoscope, the intensity pattern on the scene is non-uniform, showing a Gaussian shape at longer WDs and an annular shape at shorter ones. Due to the illumination pattern and the built-in blocking filter, it is not feasible to record the system’s IRF for FLT processing using conventional methods.

The built-in blocking filter results in only high-intensity light (the center of the Gaussian distribution) passing through the endoscope. In contrast, the peripheral areas, where light intensity gradually decreases, are blocked and attenuated to nearly zero. Consequently, only a limited number of pixels capture the IRF, while the remaining pixels record only noise. Figure 2c,d illustrate the IRF intensity image, highlighting both a location with high illumination intensity and another with attenuated light where only noise is captured. Hence, because of the NIR built-in filter, the light with lower intensity will be completely filtered, and one can see only the high-intensity illumination in the center of the scene.

Hence, one cannot uniformly excite the whole FoV. Nevertheless, due to the limited dependence of fluorescence lifetime on the intensity of the excitation, in situations where the intensity of the excitation is not uniform, the fluorescence lifetime will still exhibit a uniform lifetime image for the area under investigation.

Figure 3 shows an image of the NIR fluorescence resolution target (USAF 1951) as seen through the endoscope. At the WD of 3 cm, all the groups of the resolution target are in the FoV. The variation in the gray level in the green vertical line shows that the fourth element in group 1 is distinguishable; thus, the FLT endoscopy system’s resolution is 2.82 lp/mm. The resolution of the system is limited by the number of pixels in the sensor of 128 × 128 pixels.

## 4. Fluorescence Lifetime Processing

FLT imaging is not a direct imaging modality, and the datasets need to be post-processed to quantify fluorescence lifetime. Such postprocessing typically involves a model in which (non-) iterative optimization methods are employed to estimate the lifetime. The measurement decay is mathematically defined as the convolution of the IRF with the fluorescence decay.

Many methods exist to estimate the lifetime in the time or frequency domain. The most straightforward and fast technique is rapid lifetime determination (RLD) [29]. Its minimal form for mono-exponential decays requires only two time points (gates) in a decay to calculate its lifetime. This method needs to account for the IRF of the system, leading to low accuracy for short lifetimes compared to IRF.

Alternatively, the most accurate method to determine lifetime is re-convolution based on curve fitting, such as nonlinear least squares fitting, using the Levenberg–Marquardt minimization algorithm [30]. Although this method can consider the IRF to estimate lifetimes accurately, it is relatively slow and computationally expensive. Furthermore, the acquisition could be faster since many time points in the fluorescent decay are required. Reference [31] introduced re-convolution based on Laguerre expansion. The linear processing without any iterative optimization makes this approach less complex, and fewer time points result in shorter processing times at a reduced computational cost. However, this technique requires an initial guess for lifetime estimation and does not consider any time offsets introduced by the height variation of the imaging surface.

The frequency-domain fluorescence lifetime approach is based on the fit-free technique called the phasor approach proposed in [32,33]. This approach originates from the analysis of sinusoidal modulation signals; recently, [33] demonstrated the phasor analysis to signals generated from periodic pulsed excitations (particularly for mono-exponential decays). Since phasor analysis is a simple and fit-free technique, non-imaging experts are increasingly adopting it [34]. However, any offsets regarding changes in the WD due to the surface’s height profile directly impact the signal’s phase, resulting in estimation errors on the fluorescence lifetime.

Over the last decade, a rising interest has been in data-driven and model-free image processing techniques. For instance, machine-learning (ML) and deep learning (DL) methods have boomed and profoundly impacted various domains, such as image processing, computer vision, natural language processing, robotics, and speed recognition [35]. DL possesses high-level, robust performances because of feature hierarchical representation learning, which helps learn complex functions and high-dimensional data. Convolutional neural networks (CNNs) are the most extensively used architectures for DL. Various CNN architectures, such as AlexNet [36], GoogLeNet [37], and Microsoft ResNet [38], have introduced significant advancements in DL. DL is also widely studied for its applications in the biomedical field, particularly fluorescence signal prediction from label-free images [39], fluorescence microscopy image restoration [40], super-resolution fluorescence microscopy, and analysis of three-dimensional fluorescence images [41]. DL has emerged with promising results for fitting-free fluorescence lifetime estimation in fluorescence lifetime microscopy (FLIM) applications. In reference [42], Gang Wu. et al. demonstrated fluorescence lifetime processing based on a fully connected model for the first time, which can calculate fluorescence lifetime at least 180-fold faster than conventional methods [43]. FLI-Net and Net-FLICS architectures, respectively, based on 3D and 2D CNNs, have been proposed with promising accuracy for FLIM [43,44]. In principle, fluorescence decay data will be treated as 3D images with two spatial dimensions and one temporal dimension. 1D-ConvResNet [45], based on 1D CNN and feature extraction, uses only the temporal dimension to calculate fluorescence lifetime. In reference [45], Dong Xiao, Yu Chen, et al. have shown that in even higher dimensional CNNs, neural networks extract only temporal features, and thus, using higher dimensional CNNs not only massively increases training calculations but also requires hardware with much more computing power and large memory. By contrast, 1D CNNs are faster for the same data with 256-time points and more accessible to implement on FPGAs opening doors for hardware acceleration.

Despite the remarkable advantages of FLI-Net and 1D-ConvResNet, each time series requires a significant number of time points (more than 50) as input to the CNN, resulting in long acquisition times for a single measurement. Due to intrinsic limitations with acquisition, speed, and sensitivity, TCSPC and intensified CCDs, in most cases, are inadequate for real-time fluorescence lifetime imaging.

To achieve real-time fluorescence lifetime imaging, both acquisition time and the processing of the measurements into lifetime need to be done in real-time. Since time-domain lifetime imaging requires several time-resolved time points, the amount of time points must be kept reasonable to limit the acquisition time. The number of necessary time points influences the robustness of the processing, whereas the latter depends mainly on the sensitivity of the instrumentation.

The *tau*CAM used in this work supports high repetition rates and high NIR sensitivity, resulting in fast macroscale wide-field measurements. Moreover, limiting the number of time points during the decay, measuring them with high sensitivity with short acquisition times, and then processing them with fast and robust methods is key to achieving real-time FLT imaging.

It should be noted that, for most of those methods, recording the IRF of the system is essential before the measurement as one of the pre-defined parameters. However, as mentioned in Section 3, recording the IRF for our endoscopy setup is not possible because of the non-uniform Gaussian illumination and built-in excitation filter that significantly attenuated the excitation light, leading to an almost noisy signal for most of the pixels, as shown in Figure 3. Hence, one needs a method free of using the IRF of the system to calculate the FLT.

We demonstrated our FLTCNN architecture for macroscopic wide-field and endoscopy FLT imaging trained on fully synthetic datasets, requiring limited time points. The performance of the system was compared with the conventional curve fitting in wide-field FLT imaging and endoscopy FLT imaging. Finally, we demonstrated the potential of the system to quantify the FLT of different NIR phantoms.

### 4.1. FLTCNN Architecture Design, Training, and Validation

The schematic of FLTCNN for mono-exponential fluorescence decay analysis is shown in Figure 4. For simplicity, this study was limited to the mono-exponential model. The topology consists of two parts: first, the feature extraction block to extract general temporal features (indicated in yellow), and then the down-sampling block, followed by several convolutional, batch normalization (BN), and the nonlinear activation function rectified linear unit (ReLU) layers, for each block to extract fluorescence lifetime. The FLTCNN is inspired by FLI-Net topology with some modifications in the architecture to adopt the neural network to estimate the fluorescence lifetime of the data recorded by the *tau*CAMTM with 6-time points.

The input of the FLTCNN is an image stack consisting of spatial and temporal resolved data (128,128,6). The mono-exponential parameter (lifetime τ) is estimated for each pixel, providing the lifetime array as the exact spatial dimension as the input (128,128). The first 3D convolutional layer filters the input along the temporal dimension at each spatial dimension to avoid any feature dependency along the spatial dimensions. To achieve this, a 3D convolution with a kernel size of (1,1,6) and 64 filters is applied across the depth of the input tensor. A stride of 1 is set so that the filter moves one unit at a time in all dimensions to maximize spatial independence and temporal feature extractions. Instead of the conventional ResBlock3D, a customized one (SimiResBlock3D), where the skip connection is used to add the input (or a transformed version of it) back to the main path after the convolutional layers, is employed. SimiResBlock3D with 32 filters of (1,1,6) and stride of 1 is added, followed immediately afterwards to extract temporal information further.

The key feature of this residual block is the skip connection that adds the original input (or its transformation) back to the output of the convolutions, allowing the network to learn both the original mapping and additional ones extracted from the convolutions. This change also addresses vanishing gradients and representational bottlenecks during the network training [40]. To ultimately obtain image reconstruction of size (x × y) via a sequence of down-sampling, a transformation from 4D to 3D was required. Thus, after the SimiResBlock3D (output of x × y × kernel size × number of filters), the tensor was reshaped to dimension (x × y × (kernel size × number of filters)). After this transformation, a 2D convolutional layer of size (1,1) possessing 32 filters and 2 subsequent SimiResBlock2D couplets possessing size (1,1) was employed before the DownSampleBlock. The (1,1) size of 2D convolutional filters guarantees maintaining spatially independent feature extraction.

After extracting the common temporal features of the whole input, a sequence of four 2D convolutional layers is employed to sample the data and calculate the 2D fluorescence lifetime image. The padding for all the convolutional layers is set to be “same” to ensure that the output has the same spatial dimensions as the input (and minimize spatial dependency).

We used two features that make our CNN different from earlier works: first, the temporal dimension is set to 6-time gates, leading to accelerated network training and promising real-time FLT estimation, and second, instead of distributing the time gates over the whole decay, they are distributed throughout the decay starting at the peak value. This approach helps the CNN to mimic curve-fitting with limited time points on the decay part of the signal with fluorescence information. Timing jitter or timing differences due to differences in optical path lengths due to unevenness of the surface cause timing offsets in the recorded fluorescence signal. Conventional methods are not always robust against these timing offsets. However, recording the decay after the peak value and normalizing all the signals with different offsets to the origin of the coordinate mitigates the impact of the offset on the FLT estimation in our approach.

The proposed FLTCNN was developed utilizing TensorFlow 2.14 on Google Colab. The loss function for training is a mean square error (MSE) as illustrated in (1):(1)$$L\left(\Theta \right)=\frac{1}{M}\;\overset{M}{\underset{i=1}{\sum }}\|F\left(Y_{i},\Theta \right)-Y_{i}^{\prime }{\|}_{2}^{2}$$
where *F*(.) is the end-to-end mapping function of training parameter Θ, *Y* is the input, $Y_{i}^{\prime }$ is the corresponding ground truth value of the *i*th branch, and *M* is the batch size.

The optimizer used the root mean square propagation (RMSprop) algorithm with a decrementing learning rate from 0.1 and the rate of $e^{-0.1}$ discrimination after every 10 epochs to achieve faster convergence, prevent oscillations, and avoid becoming stuck in undesirable local minima. Early stopping of 32 patient epochs was scheduled to prevent overfitting. Other hyperparameters, such as the kernel size, the stride of different convolutional layers, and the number of layers, were optimized based on exhaustive trial and error. The training dataset contains 4000 images, split into training (3200) and validation (800) datasets. The hyperparameters are optimized based on validation sets, and the model is assessed by a new batch of synthetic data with the same parameters as the training sets, and real FLT data as the test sets. It should be noted that, since the synthetic data is generated with the same formula and does not contain any real data, we do not expect any bias in any folds (if the validation set splits in for instance “n” folds), because of which hyperparameters are optimized based on only validation sets. The batch size of the training was chosen to 16 and the training epochs were set to 100.

### 4.2. Synthetic Data for Training

To obtain large training datasets to train FLTCNN and ensure that the architecture is robust and able to extract lifetime efficiently and accurately, 4000 images were generated using MATLAB R2020b. Each pixel was simulated utilizing the modified National Institute of Standards and Technology (MNIST) database to acquire a spatial map of 128 × 128 pixels along generated fluorescence decays at each non-zero pixel using a mono-exponential model convolved with synthetic IRF illustrated in (2):(2)$$D\left(t\right)=IRF\left(t\right)*A.e^{-\frac{t}{\tau }}+\varepsilon \left(t\right)$$
where *A* and, *τ* are the pre-exponential and lifetime of the fluorescence signal, respectively, and ε(*t*) is the Poisson noise.

The IRF of the FLT system is the convolution of the laser pulse with the gate window of the sensor. Since the pulse width of the laser used in this research (<35 ps) is significantly shorter than the gate width of the sensor (4 ± 0.017 ns) [28], the contribution of the laser pulse to the IRF is negligible.

The IRF was modeled as a super-Gaussian [46,47] at each pixel illustrated in (3):(3)$$IRF\left(t\right)=\exp[-2{\left(\frac{2\left(t-t_{0}\right){\left(0.5\ln2\right)}^{\frac{1}{2N}}}{FWHM}\right)}^{2N}]$$
where *t* is time, *t*_0_ is the Gaussian curve center, *N* is the Gaussian order, and *FWHM* is the full width at half maximum of the gate. The *FWHM* was randomly chosen in the range of (4~4.5) ns for each pixel for the robustness of the CNN training. *N* is set to be 4 such that it approximately matches the actual measured profiles. Since all the parameters of fluorescence decay are well understood, using synthetic data to train the CNN is a feasible approach. Additionally, [28] demonstrated that the IRF of the *tau*CAM remains stable across different settings. This results in a set of 16,384 different IRFs. The synthetic fluorescence decay (SFD) was generated using (2) and (3) for each spatial pixel location. Figure 5 shows the synthetic SFD generation flow.

During the generation of each SFD, the mono-exponential decay was convolved with one of the IRFs taken from the generated set and normalized to its maximum value. A was set to A~U [20, 100] (U [] denotes uniform distribution), followed by adding Poisson noise to mimic the background and shot noise to SFD. The signal-to-noise ratio of the fluorescence decay is the square root of the total photon count:(4)$$SNR=\sqrt{\overset{n}{\underset{t=1}{\sum }}D\left(t\right)}$$
where *n* denotes the number of time bins.

Then, each SFD was normalized again, ensuring the chances of low and high SNR signals for training the FLTCNN are the same. Finally, each signal was sampled starting at its maximum value through the decay with the required 6-time points. The 6-time points were evenly spaced, starting from the peak through the decay. The lifetime τ for each image was varied over the range, t~U [0.6,2] ns corresponding to values typically encountered in these applications. As mentioned in [45], limiting the range of lifetimes can increase the precision of the lifetime extraction. Hence, we kept the lifetime range between the mentioned interval as the typical NIR fluorescence decay must reach high precision and accuracy and less mean absolute error (MAE).

Table 1 compares architectures between conventional CNNs and feed-forward dense neural networks (FFDNN). In contrast to the training of 2D/3D DNNs, which typically require several hours due to many parameters, the FLTCNN features a significantly lower count of trainable parameters, leading to reducing the necessary memory and computational resources for floating-point operations. This is because, while proposed architectures demand input data with at least 200-time bins, the FLTCNN functions efficiently with data from only 6-time bins, significantly reducing the input size in the temporal dimension. As a result, the FLTCNN needs a less complicated architecture with fewer parameters, leading to a substantially shorter period of tens of minutes to train. It should be noted that architectures such as the ELM, 1D-ConvResNet, and FLAN may have shorter training times than the FLTCNN; however, these handle 1D histograms with 256-time bins. Therefore, the relatively brief training duration of the FLTCNN, with its 3D tensor input, denotes significant efficiency. This advantage of shorter training times is beneficial as it allows for rapid re-training and application of the CNN across various FLT uses. A more critical parameter is the inference times of the architecture used to estimate FLT data, especially for real-time applications. The FLTCNN can calculate an FLT image of 128 × 128 in 0.9 s processed on a GPU. Unfortunately, there are not any reports about the inference times of other architectures.

## 5. Network Evaluation

To assess the robustness of the FLTCNN model, Figure 6a presents the training and validation mean absolute error (MAE) curves over 100 epochs, demonstrating the model’s excellent stability in convergence to the minimal loss level. Additional evaluations used a newly created synthetic dataset previously unseen by the CNN, which matched the training parameters as a test dataset. Figure 6b provides a boxplot detailing the MAE across the test datasets. This dataset, split into five subsets, includes 100 images per subset with SNR ranging from 10 dB to 25 dB, and the lifetime range is maintained consistent with the training set. As depicted in Figure 6b, the average MAEs for lifetime estimation decrease notably as the SNR improves, illustrating enhanced precision.

Conversely, datasets characterized by lower SNR exhibit broader error distributions, expecting the improved performance of FLTCNN with increasing SNR. It should be noted that the influence of the IRF on lifetime estimations is implicitly accounted for within the model, consistent without the need for re-convolution. The model is trained across various IRF FWHM values to ensure robust performance.

Quantitative analysis of lifetime MAE under various conditions further illustrates the resolving capability of FLTCNN. The test data spanned over the lifetime range of 0.2~3 ns, aiming to probe the model’s performance beyond the conditions of the training set. Figure 6c illustrates outcomes for SNR values spanning 30~400 (A = 10, 20, 50, 100), where MAE values increase with lower SNR, consistent with prior findings. The model’s capability to resolve lifetimes is contingent upon the range of lifetimes included in the training data, showing a linear increase in MAE when extending beyond the training range, as demonstrated in [45]. Thus, the resolving capability of FLTCNN can be tuned based on the selected training data parameters.

To assess the robustness and efficacy of the feature extraction within the main block (highlighted in yellow in Figure 4), the outputs from the final activation layer of this block were captured while processing 1000 newly simulated SFD data voxels, which were not previously utilized in training or validation. These high-dimensional features were subsequently flattened and transformed into a three-dimensional feature space using t-distributed stochastic neighbor embedding (t-SNE). The resulting distribution is depicted in Figure 6d as a scatter plot, with colors assigned based on the lifetimes’ ground truth (GT) values. The observed continuous gradient in color within clusters in the 3D t-SNE plot against the simulated lifetimes (τ~U [1,2] ns) demonstrates effective and sensitive feature extraction capabilities for lifetime-based parameter estimation and highlights variability within the group.

Further evolutions were conducted based on real data recorded by the *tau*CAM. An ICG uniform phantom (excitation/emission peak: 789 nm/814 nm [19]) was imaged in the wide-field macroscopic regime (with imaging lens), and the recorded data were processed with the curve-fitting (Lmfit) method. Validation of the *tau*CAM with the curve-fitting approach was already proven with a Mini-tau fluorescence lifetime spectrometer (Edinburgh Instruments) for different fluorescence dyes with different excitation/emission wavelengths [53]. The recorded data were first processed with Lmfit, and then the same data were fed into the FLTCNN.

The estimated average fluorescence lifetime is 1.14 ns and 1.11 ns, by Lmfit and FLTCNN, respectively. Figure 7a–c shows the FLT and intensity images of the uniform ICG phantom. The illumination is intentionally made of non-uniform to test the performance of the FLTCNN and estimate its lifetime.

Unlike the FLT images, which display good uniformity, the fluorescence intensity image exhibits non-uniformity. This highlights that FLT does not depend on the illumination intensity. Figure 7d illustrates the FLT histogram of both methods, which almost overlap each other as expected.

To confirm the FLTCNN’s performance with the FLT endoscopy system, the ICG uniform phantom has been imaged in a wide-field endoscopy regime by the endoscopy system. As shown in Figure 8, the FLT image shows good uniformity, and the average fluorescence lifetime is estimated at 1.18 ns, which agrees with the previous analysis.

The efficacy of the FLTCNN in handling height variations was validated. This experiment angled a uniform phantom relative to the horizontal axis with 45° beneath the endoscope. As discussed in Section 5, traditional RLD and Phasor methods fail to estimate lifetime under height variations accurately. In contrast, the FLTCNN model demonstrates insensitivity to such height differences, consistently predicting the correct lifetime. Utilizing the same uniform ICG phantom as in previous analyses ensures that the expected lifetime remains consistent. Figure 9 displays the intensity and FLT images of the angled uniform phantom, where the FLT image maintains good uniformity with an average lifetime that confirms previous results. Thus, FLTCNN demonstrates some robustness in predicting lifetime regardless of changes in the setup. Table 2 quantifies the previous analysis’s average lifetime and standard deviation (SD).

## 6. Results and Discussions

The potential of FLTCNN for endoscopy imaging was also tested on different phantoms against Lmfit. The proposed CNN was employed to analyze additional NIR FLT endoscopy images. The FLT images were acquired by the FLT endoscopy system discussed in Section 3. The gate width of the *tau*CAM was set to 4 ns, and each measured decay contains 6-time bins. Full-decay Lmfit curve fitting in the wide-field microscopic regime (utilizing a camera lens) was the reference for these experiments.

As noted in Section 3, recording the IRF of the endoscopy system is not feasible. Instead, one can employ synthetic IRFs to process the endoscopy data using Lmfit curve fitting. However, in the endoscopy regime, only 6-time points from the peak value are recorded; consequently, if the synthetic IRF is recorded similarly (with 6-time points), the resulting signal has only one sample (resembles a Dirac function). Thus, only the offset corresponding to the IRF would influence the fitting process. An alternative approach with the decay with 6-time points is tail-fitting on the fluorescence decay tail, where the IRF’s impact has diminished. It is important to note that tail-fitting on the fluorescence decay without considering the IRF differs from RLD; in this method, the fitting algorithm is applied to the decay tail, whereas RLD is used throughout the decay where the SNR is adequate (the integration gates would be positioned closely after the peak of the decay). Hence, the data recorded by the endoscopy system can be processed without considering the IRF. In the subsequent analysis, all experimental data were processed in the wide-field microscopic regime using full-decay Lmfit and wide-field endoscopy regime using 6-time points tail-fitting and FLTCNN.

The first experiment was conducted on the ICG endoscopy concentration phantom from Quel Imaging. This phantom contains five wells with different concentrations of ICG. Given that all wells contain the same dye, the lifetime distribution of the lower concentrations should ideally match that of the 1000 nM concentration. Figure 10a–c presents FLT images estimated by Lmfit full-decay in the macroscopic regime, Lmfit with 6-time points, and FLTCNN in the wide-field endoscopy regime. As shown in Figure 10a, although the camera sees almost all wells, Lmfit full decay analysis accurately estimates only the first 3 wells. Additionally, as concentration decreases, the average lifetime, which is expected to remain stable, shifts to a shorter lifetime (as depicted in the histogram in Figure 10e). Lmfit with 6-time points, however, correctly estimated only the lifetime at the highest concentration where the SNR is high. Figure 10b demonstrates that, with decreasing concentration, the average lifetime shifts to longer values, and the standard deviation increases, which means the histogram broadens; this could happen due to the low SNR of the signal. However, FLTCNN analysis of the same 6-time points shows a similar average lifetime and uniform distribution for the first three wells. Yet, the average lifetime for the well with a 10 nM concentration deviates significantly from the others. The number of outliers has notably increased due to the lower SNR (fewer photons), consistent with previous analyses. Table 3 quantifies the average lifetime and SD of the different concentration wells.

We further evaluated the performance of the FLT endoscopy system using three differently shaped ICG phantoms: distortion, coin, and vessel from Quel Imaging, all with a concentration of 1000 nM, to evaluate the ability of the FLTCNN not to distort the spatial features of different objects of the same phantoms. Given the uniform concentration of ICG across all phantoms, a similar lifetime is expected.

As shown in Figure 11, the FLT remains consistent and uniform across all cases, with overlapping distributions, which are anticipated due to the homogeneity and continuity of the phantoms and are confirmed by Lmfit full-decay analysis (Figure 11a–c). Comparing the distortion phantom to the coin and vessel phantoms, the lifetime distribution for the distortion ICG phantom slightly extends to longer lifetimes.

This extension is attributed to a significant increase in outliers at the edges of the FoV where excitation intensity is low (fewer photons). Figure 12 illustrates the lifetime distribution of the ICG phantoms as estimated by Lmfit with 6-time points. There is a significant increase in outliers, and the average lifetime deviates notably compared to FLTCNN and Lmfit full decay.

Conversely, FLTCNN provides a more localized distribution. Despite the non-uniform fluorescence intensity, the FLT images remain almost uniform, demonstrating the robustness of FLT against non-uniformity. Furthermore, the system should not distort the spatial features of the objects; hence, FLTCNN is designed to be spatially independent.

The average lifetime and standard deviation values for each phantom and method are quantified in Table 4, showing FLTCNN consistently outperforms Lmfit with 6-time points, delivering reliable results with fewer time points.

In the third experiment, we used a mixed phantom, Quel Imaging, which contains ICG and an OTL38-mimicking dye in their standard substrate and with substrate mimicking lung tissue.

As seen in the fluorescence intensity image in Figure 13, each well lights up due to the fluorescence emission spectral overlap, making it impossible to distinguish the fluorophores in the different wells. However, the FLT image reveals distinct lifetimes for each well, making identification possible. By comparing the average lifetime and SD of each well using Lmfit as GT, it was demonstrated that FLTCNN demonstrates high performance and robustness, successfully reconstructing FLT images with only 6-time points.

In contrast, Lmfit with 6-time points shows significant deviation from the GT lifetimes, failing to estimate the correct lifetime. Table 5 quantifies all wells’ average lifetime and SD using different methods. Notably, the CNN enables multiplexing independent of fluorescence intensity, allowing it to have good precision of the different lifetimes of various phantoms.

## 7. Conclusions

In this study, we demonstrated the capability of the *tau*CAM with an endoscope for FLT imaging. Our work explored the application of this advanced imaging technology in endoscopic settings, highlighting its potential to achieve high-speed NIR fluorescence lifetime measurements. The *tau*CAM showed its potential capabilities in NIR FLT imaging; integrating the *tau*CAM with the NIR/ICG Hopkins endoscope allowed us to explore the system’s possible potential for enhancing real-time FLT endoscopic imaging as a minimally invasive diagnostic tool.

A significant aspect of this research was incorporating a CNN architecture, specifically the FLTCNN, to process FLT data without the real IRF of the system since a new IRF needs to be measured for each measurement for conventional processing algorithms. The FLTCNN demonstrated robustness and efficiency, reducing computational load compared to traditional curve-fitting techniques. This model processed data with acceptable accuracy, effectively handling experimental data even in challenging conditions, such as non-uniform intensity and height differences. The FLTCNN can deliver reliable lifetime estimations with fewer time points (6-time points) within 0.9 s per image (128 × 128) using GPU. This processing speed underscores its potential for real-time applications in biomedical imaging.

The experiments with uniform ICG phantom, where the illumination was non-uniform, and the concentration ICG phantom with different concentrations of fluorophores highlighted the *tau*CAM’s capability with FLTCNN to produce uniform FLT images, where the system maintained consistent lifetime values across different concentrations, demonstrating its robustness. Further, the mixed phantoms containing ICG and OTL38 in lung and standard material comprehensively evaluated the system’s performance. It is challenging to distinguish different wells based on intensity alone. However, the FLT images revealed distinct lifetimes for each well, effectively differentiating between the different fluorescent dyes. This capability is crucial for practical applications, where distinguishing between malignant and benign tissues based on their FLT characteristics can significantly enhance diagnostic accuracy.

Future work will further optimize the endoscopy system combined with *tau*CAM and validate its performance in clinical settings. Furthermore, re-designing the architecture with 1D CNN might promise faster training and lifetime estimation, less memory, and simple hardware to implement. A key development area will be the addition of RGB imaging capabilities, allowing for the overlay of FLT images on top of RGB images to provide more comprehensive visual information.

Additionally, achieving real-time measurement with a high frame rate per second remains a significant challenge that must be addressed in future developments. The transition to real-time imaging will involve overcoming technical hurdles related to data acquisition speed and processing efficiency. Advances in data processing algorithms will be critical in achieving this goal, paving the way for real-time FLT endoscopy imaging in clinical practice.

In conclusion, integrating the *tau*CAM with the endoscope and employing advanced data processing techniques, particularly the FLTCNN, represents the potential of FLT endoscopy imaging with the hope of advancing in this field. This study has laid the groundwork for further exploration and optimization, highlighting the potential of this technology to make substantial contributions to medical diagnostics and surgery. By enhancing the ability to differentiate between malignant and benign tissues, this technology promises to improve diagnostic accuracy and patient outcomes in the future.

## Figures and Tables

**Figure 1 sensors-25-00450-f001:**
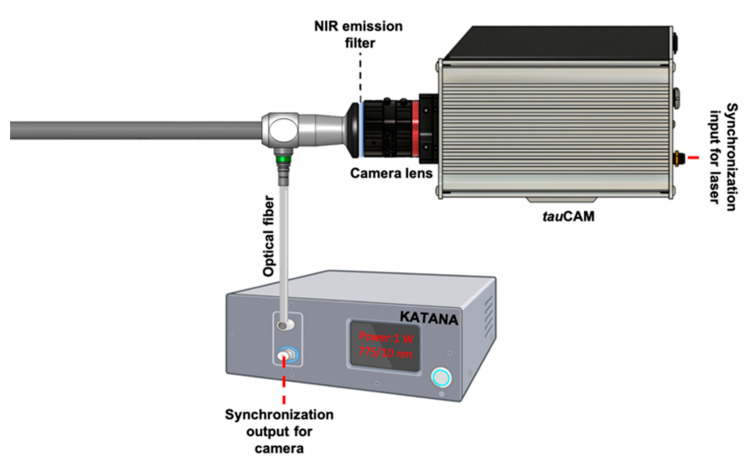
Schematic diagram of the FLT endoscopy imaging.

**Figure 2 sensors-25-00450-f002:**
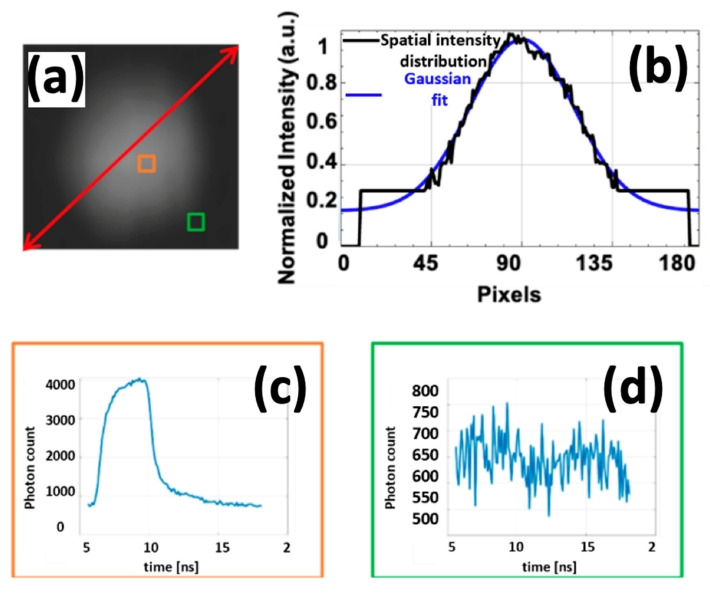
(**a**) Intensity pattern of the endoscope illumination through the FoV at WD of 3 cm, and (**b**) the 1D pattern through the mentioned red line in the diagonal direction, which has a Gaussian distribution. (**c**) The orange square marks the location of strong illumination, with its corresponding IRF showing a clear signal. (**d**) The green square indicates a region where the endoscope attenuates the light intensity, corresponding to a noise-dominated IRF.

**Figure 3 sensors-25-00450-f003:**
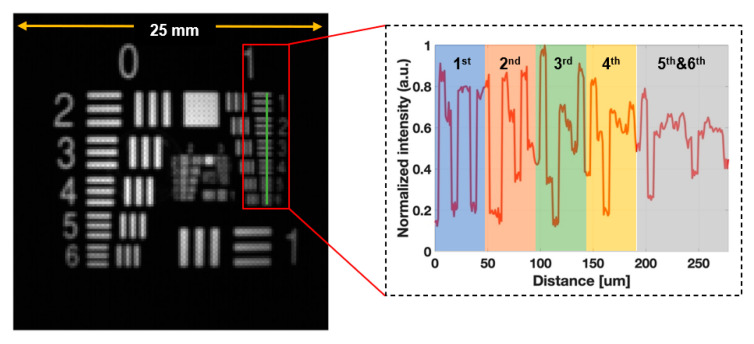
Resolution analysis of the FLT endoscopy system based on resolution test target USAF 1951.

**Figure 4 sensors-25-00450-f004:**
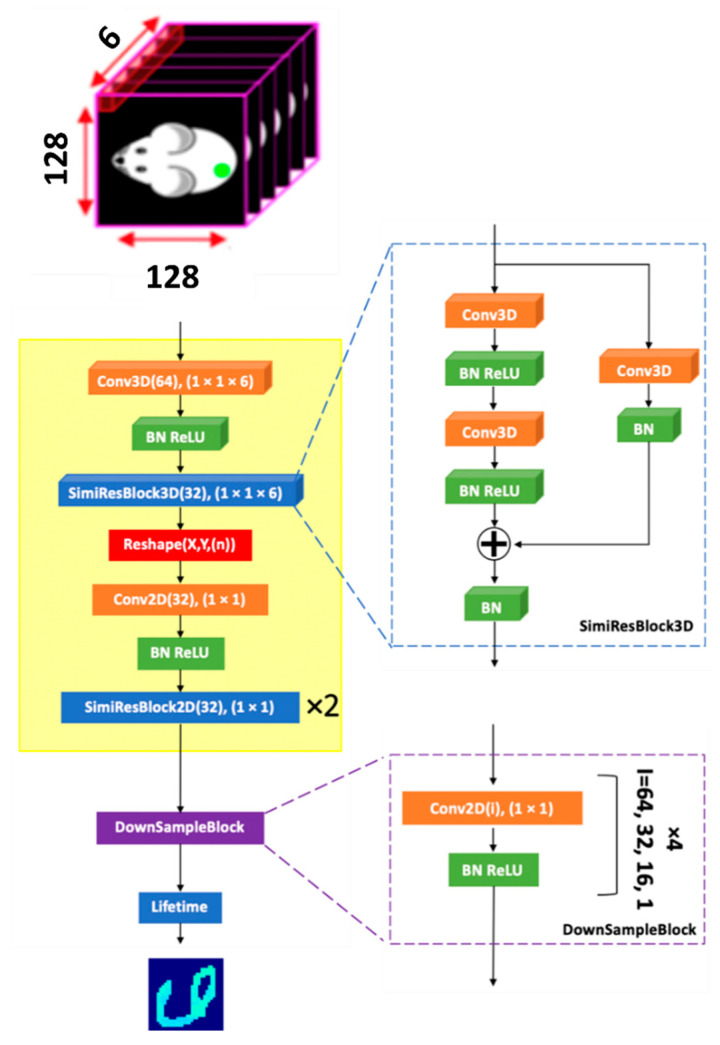
Topology of FLTCNN to analyze mono-exponential fluorescence decays. The details of hyperparameters in each layer in parenthesis represent the number of filters, and the kernel size, respectively. The input is an image stack of (128,128,6). The architecture of SimiResBlock, and DownSampleBlock (consists of 4, 2D convolutional layers with decrementing filter sizes) are shown with a dashed box. The BN and the ReLU are added after convolutional layers.

**Figure 5 sensors-25-00450-f005:**
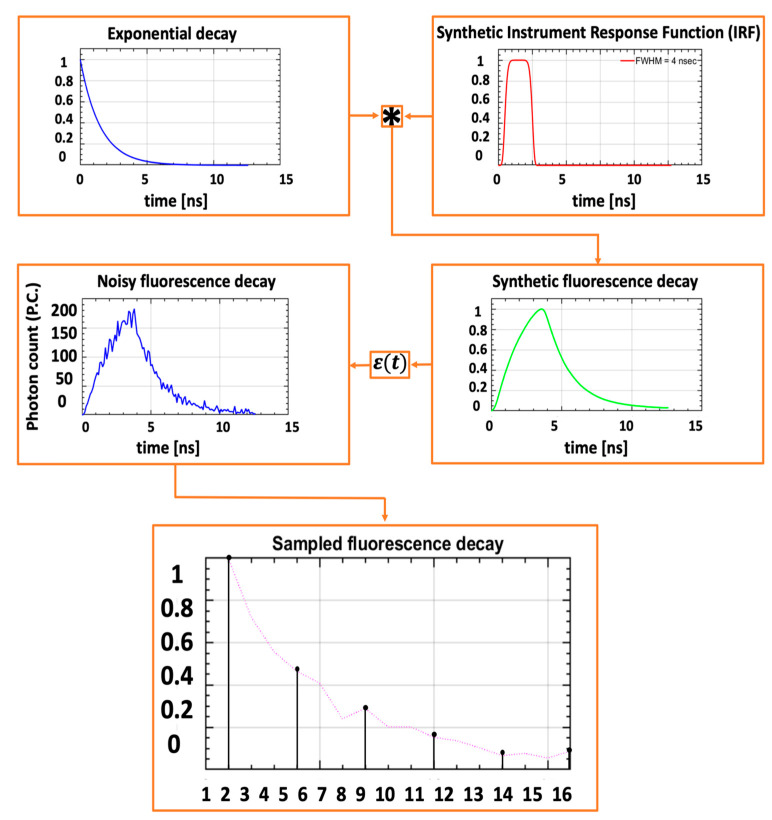
Synthetic training data generation flow for mono-exponential fluorescence signal model.

**Figure 6 sensors-25-00450-f006:**
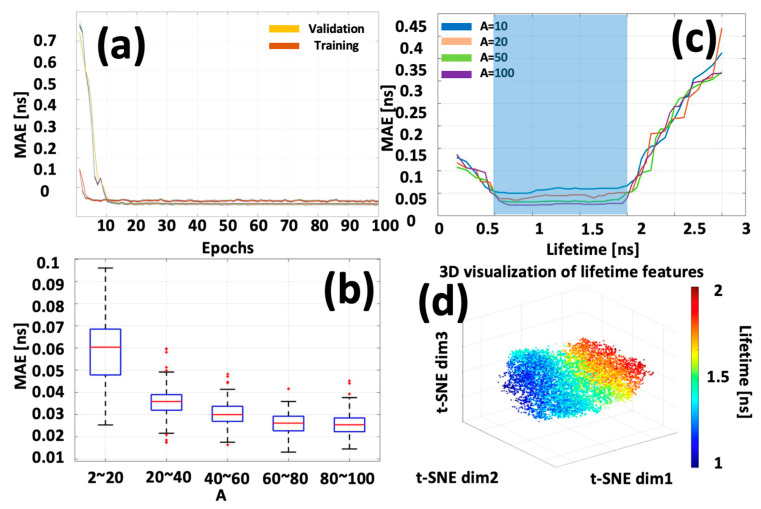
(**a**) MAE graph of training/validation vs. epochs. (**b**) The MAE of predicted results of the testing datasets. (**c**) The mean value of MAE for a lifetime is under different conditions. The SNR takes the value between 20 to 1000 for A = 10, 20, 50, and 100. The blue area denotes the lifetime range of training data. (**d**) t-SNE visualization was obtained via the last activation map before the down-sampling block, where each point represented a TPSF voxel and was assigned a randomized lifetime value.

**Figure 7 sensors-25-00450-f007:**
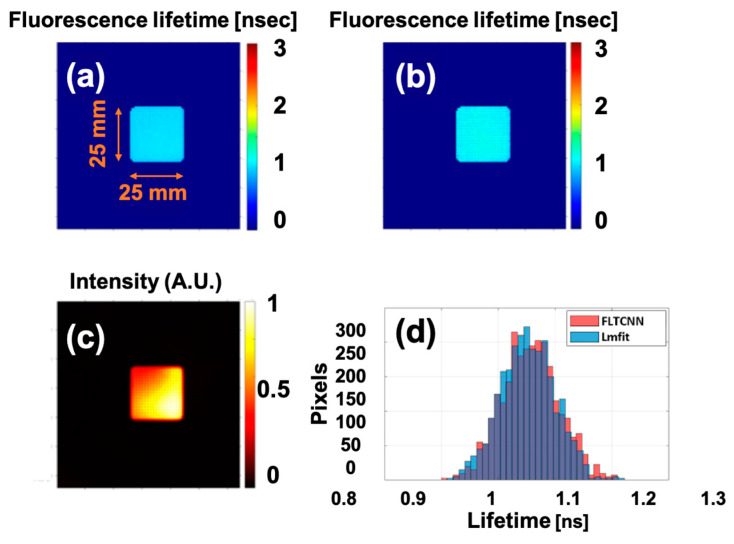
FLT image of a uniform ICG-equivalent phantom predicted by (**a**) Lmfit (Levenberg–Marquardt), (**b**) FLTCNN in the macroscopic wide-field regime. (**c**) Shows the normalized fluorescence intensity of the phantom. (**d**) FLT histogram of the ICG uniform phantom processed with Lmfit and FLTCNN.

**Figure 8 sensors-25-00450-f008:**
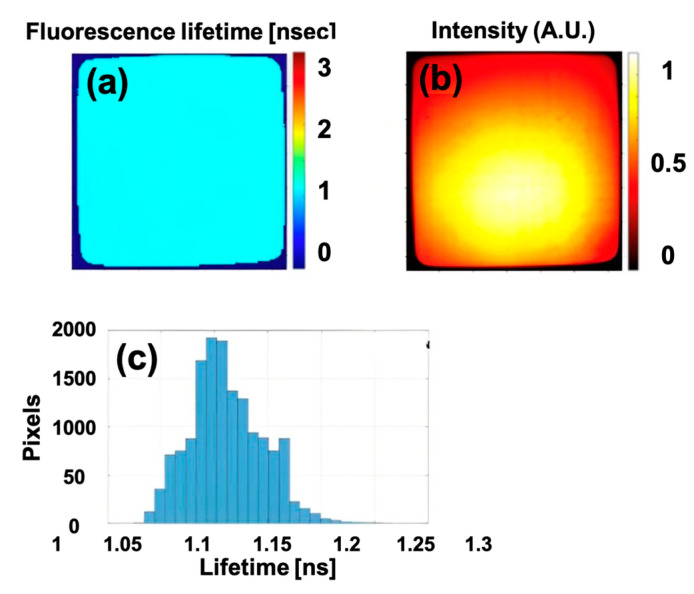
(**a**,**b**) FLT and intensity images, and (**c**) FLT histogram of the ICG uniform phantom captured by FLT endoscopy system and processed with FLTCNN algorithm.

**Figure 9 sensors-25-00450-f009:**
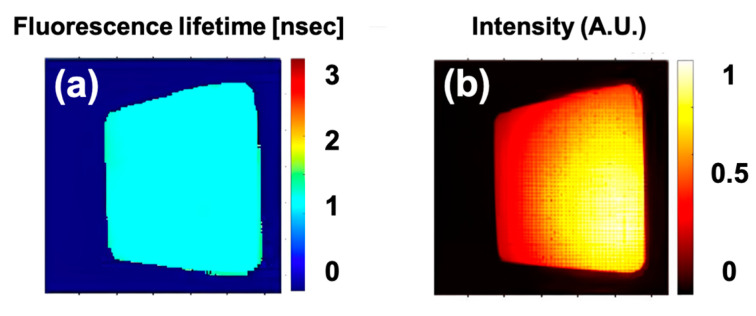
(**a**,**b**) FLT and intensity images of the uniform ICG phantom under an angle.

**Figure 10 sensors-25-00450-f010:**
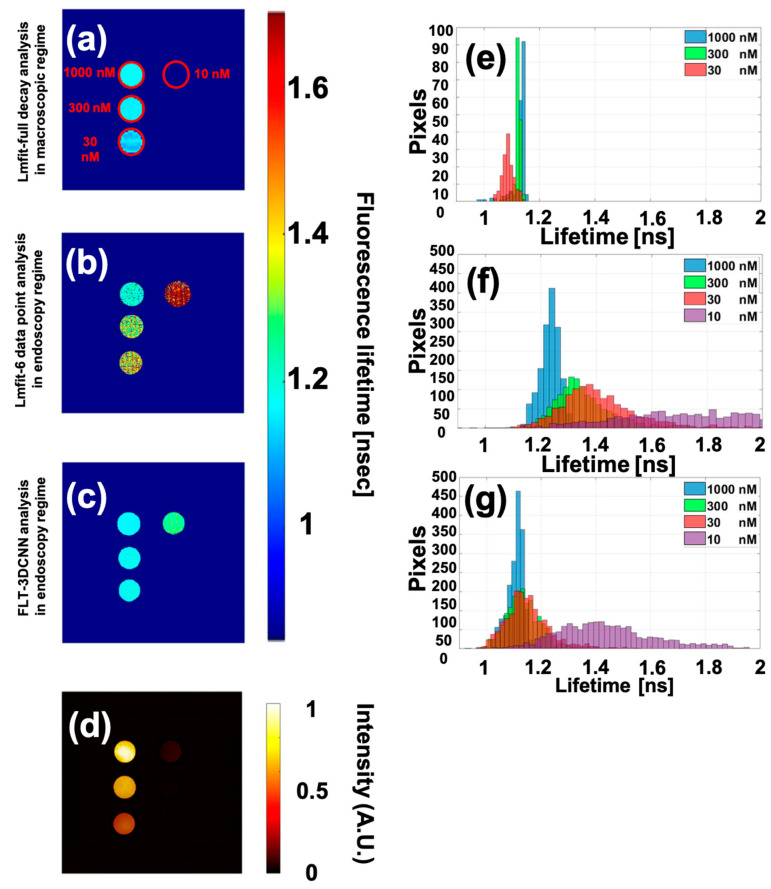
(**a**) Lmfit analysis in the macroscopic regime, (**b**) Lmfit analysis of 6-time points in the endoscopy regime, (**c**) FLTCNN analysis in the endoscopy regime, and (**d**) fluorescence intensity image of the concentration ICG phantom, Quel Imaging. (**e**–**g**) histogram of each well related to each approach to predict lifetime.

**Figure 11 sensors-25-00450-f011:**
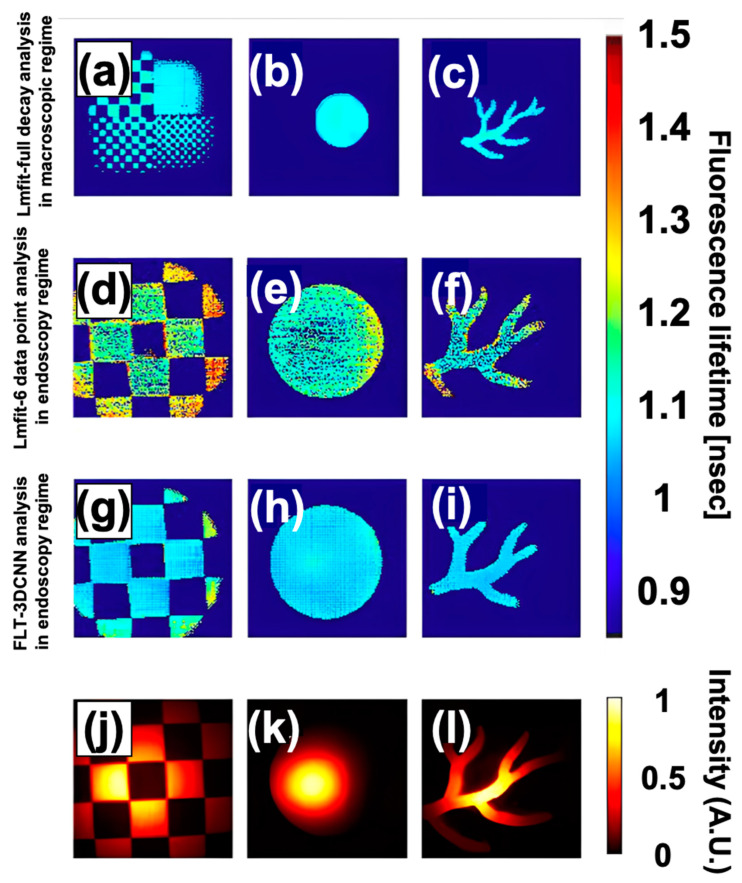
FLT images of ICG-equivalent phantoms (distortion, coin, and vessel, Quel Imaging) analyzed with (**a**–**c**) Lmfit full decay, (**d**–**f**) Lmfit 6-data point, (**g**–**i**) FLTCNN, and (**j**–**l**) fluorescence intensity.

**Figure 12 sensors-25-00450-f012:**
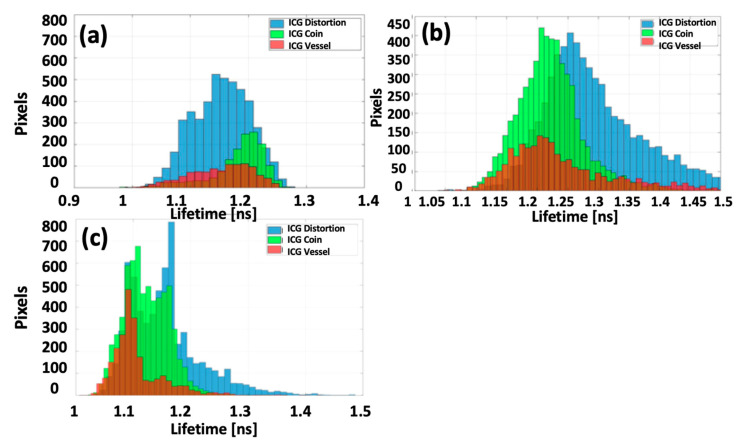
The lifetime distribution of the ICG distortion, coin, and vessel phantoms calculated by (**a**) Lmfit analysis in the macroscopic regime, (**b**) Lmfit analysis of 6-time points in the endoscopy regime, and (**c**) FLTCNN analysis in the endoscopy regime.

**Figure 13 sensors-25-00450-f013:**
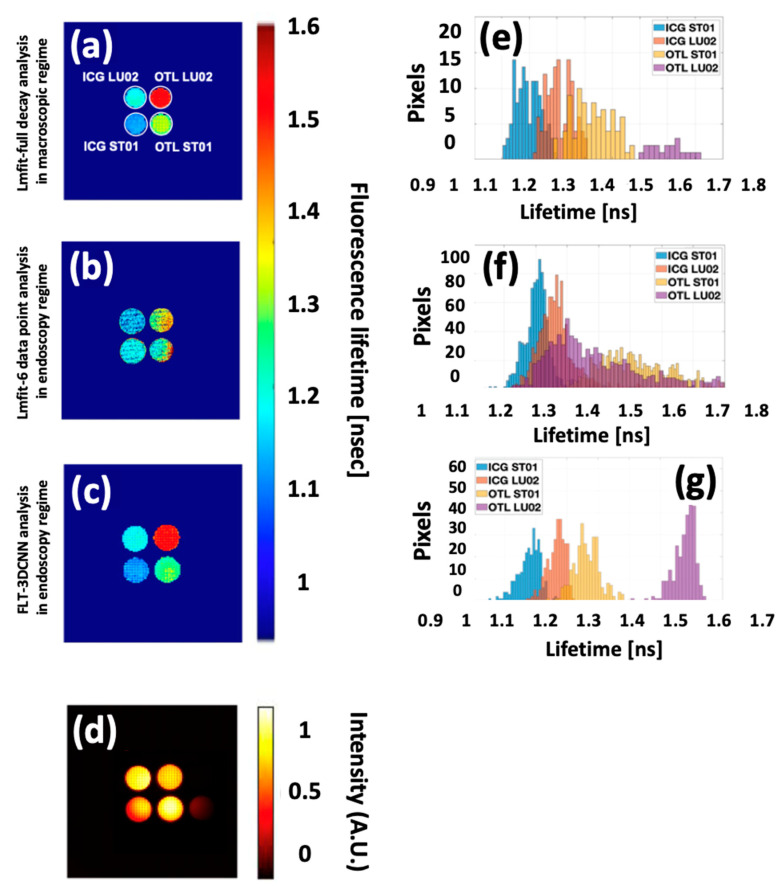
(**a**) Lmfit analysis in the macroscopic regime, (**b**) Lmfit analysis of 6-time points in the endoscopy regime, (**c**) FLTCNN analysis in the endoscopy regime, and (**d**) fluorescence intensity image of QUEL mixed phantoms containing ICG in ST01/LU02 and OTL38 in ST01/LU02. (**e**–**g**) histogram of each well related to each approach to predict lifetime.

**Table 1 sensors-25-00450-t001:** Comparison of deep learning architectures for fluorescence lifetime imaging.

	Input Data Type	# of Parameters	# of Time-Resolved Time Points	Training Platform	Training Time
FLTCNN	3D tensor	71,461	6	GPU	30 min
FLINET [43]	3D tensor	1,084,045	256	GPU	4 h
Net-FLICS [44]	2D tensor	---	256	GPU	4.5 h
QCNN [48]	1D histogram	4248	300	GPU	17 min
1D-ConvResNet [45]	1D histogram	48,675	256	CPU	23 min
DENSE NET [42]	1D histogram	149,252	256	GPU	4 h
flimGANE [49]	1D histogram	143,528	256	GPU	6.9 h
ELM [50]	1D histogram	205,600	256	CPU	10.85 s
FLAN [51]	1D histogram	23,003	256	GPU	20 min
FILM-MLP-Mixer [52]	1D histogram	26,081	256	CPU	30 min

**Table 2 sensors-25-00450-t002:** Average lifetime and SD of the ICG uniform phantom in wide-field and endoscopy system.

Uniform ICG Phantom	Method	Average Lifetime [ns]
Macroscopic regime	Lmfit	1.14 ± 0.19
Macroscopic regime	FLTCNN	1.11 ± 0.25
Endoscopy	FLTCNN	1.18 ± 0.08
Endoscopy (under an angle)	FLTCNN	1.15 ± 0.11

**Table 3 sensors-25-00450-t003:** Average lifetime and SD of ICG concentration endoscope phantom with different processing approaches.

Concentration [nM]	Average Lifetime [ns] Complete Decay (Lmfit)	Average Lifetime [ns] 6-Time Points (Lmfit)	Average Lifetime [ns] (FLTCNN)
1000	1.11 ± 0.09	1.24 ± 0.25	1.10 ± 0.23
300	1.12 ± 0.09	1.35 ± 0.27	1.13 ± 0.23
30	1.01 ± 0.13	1.42 ± 0.28	1.15 ± 0.23
10	---	1.91 ± 0.40	1.43 ± 0.29

**Table 4 sensors-25-00450-t004:** Average lifetime and SD of ICG phantom with 1000 nM with different processing approaches.

Phantom	Average Lifetime [ns] Complete Decay (Lmfit)	Average Lifetime [ns] 6-Data Points (Lmfit)	Average Lifetime [ns] (FLTCNN)
Distortion	1.15 ± 0.26	1.30 ± 0.11	1.12 ± 0.14
Coin	1.18 ± 0.26	1.37 ± 0.32	1.12 ± 0.24
Vessels	1.15 ± 0.22	1.31 ± 0.21	1.10 ± 0.20

**Table 5 sensors-25-00450-t005:** Average lifetime and SD of mixed phantom with different processing approaches.

Phantom	Average Lifetime [ns] Complete Decay (Lmfit)	Average Lifetime [ns] 6-Time Points (Lmfit)	Average Lifetime [ns] (FLTCNN)
ICG ST01	1.09 ± 0.09	1.26 ± 0.08	1.10 ± 0.24
ICG LU02	1.19 ± 0.15	1.32 ± 0.14	1.20 ± 0.28
OTL ST01	1.50 ± 0.11	1.5 ± 0.19	1.45 ± 0.26
OTL LU02	1.65 ± 0.14	1.7 ± 0.28	1.61 ± 0.33

## Data Availability

Data available on request due to restrictions.

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
