# Peer review of "Fluorescence Lifetime Endoscopy with a Nanosecond Time-Gated CAPS Camera with IRF-Free Deep Learning Method"

_sensors, 2025, doi:10.3390/s25020450_

Round 1

Reviewer 1 Report

Comments and Suggestions for Authors

The authors present their original work demonstrating fluorescence lifetime endoscopy with an analysis method that does not depend on measurement of the instrument response function (IRF). The main advancements are the use of the tauCAM detector and a deep-learning based method for predicting the lifetimes that is efficient and robust, even in the absence of the ability to experimentally measure the IRF. The theory is thoroughly presented and experiments with wide-field and endoscope-based measurements are conducted on samples of increasing complexity. The experiments are well designed, and the results align well with the theory. The work is targeted towards the goal of real-time fluorescence lifetime measurements with the endoscopy probe although the results do not quite achieve that yet.  Overall, the research is clear and compelling and only minor suggestions for improvement are noted below.

General comments:

- The authors refer to fluorescence lifetime imaging by the acronym "FLT imaging".  This reviewer is not certain that is the standard acronym in the field. The acronym "FLIM" was originally coined by J. Lakowicz in his pioneering work.  In more recent literature some refer to FLIM for fluorescence lifetime imaging microscopy (the technique which builds the image point-by-point). Please make sure the reference to FLIM is in the paper to ensure that this work is aligned with the field.

- The introduction gives a lot of references regarding basic fluorescence lifetime.  Please emphasize more relevant literature in NIR widefield approaches which is the direct competition of this method.

- The authors indicate real-time analysis of fluorescence lifetimes during endoscopy is one of the drivers for this work however the authors never present the time it time it takes to process an endoscopy acquired lifetime image with this method as currently presented so a reader can judge how close things are at present to "real-time".

Specific Comments:

- The sentence on Line 62 "limit unnecessary biopsies" needs a reference.

- When describing the tauCAM (Lines 88-94), please provide a description of the spectral response in addition to the quantum efficiency at 780 nm.  Is it relatively flat from 700-900?

- Figure 1 is not necessary in my opinion.  The only useful pieces of information from Figure 1 are the dimensions of the tauCAM.  Consider adding this Figure 2 and eliminating Figure 1.  Additionally, a picture of the fiber bundle showing where the illumination fibers are in relation to the fluorescence collection fibers would be helpful.  Consider adding this to Figure 2 or as a separate figure.

- Line 104 refers to the customized SMA adapter.  Please describe at least at a high level what the customization was.

- IRF is never explicitly defined.

- Section 4, Lines 165- 181.  This section would read better if the methods for determining lifetime were split into paragraphs on time domain and frequency domain.  As written, it is confusing.

- Line 265 - 271 seems to have a repetitive statement.  Check for correctness.

- How are the 6 time points used for the training data spaced? It's clear that they start at the decay peak, but are they evenly spaced all the way to the tail? 

- Table 4 is referenced on Line 561, however, this indicates that Table 4 has the summary of the results for the ICG phantoms with different shapes.  Table 4 is then again referenced on line 573 as correctly containing the summary results from the mixed dye phantoms.  is it possible there's a table missing with the summary results from the ICG phantoms with different shapes?

Author Response

GENERAL COMMENTS

Comment 1: The authors refer to fluorescence lifetime imaging by the acronym "FLT imaging".  This reviewer is not certain that is the standard acronym in the field. The acronym "FLIM" was originally coined by J. Lakowicz in his pioneering work.  In more recent literature some refer to FLIM for fluorescence lifetime imaging microscopy (the technique which builds the image point-by-point). Please make sure the reference to FLIM is in the paper to ensure that this work is aligned with the field.

Response 1: The acronym "FLIM" refers to Fluorescence Lifetime Microscopic Imaging; however, there is no standard acronym for macroscale fluorescence lifetime imaging.

----------------------------------------------------------------------------------------------------------------

Comment 2: The introduction gives a lot of references regarding basic fluorescence lifetime.  Please emphasize more relevant literature in NIR widefield approaches which is the direct competition of this method.

Response 2: Unfortunately, there is a lack of NIR wide-field FLT imaging. It is a very novel approach, and we are one of the pioneers in this field in the world. 

----------------------------------------------------------------------------------------------------------------

Comment 3: The authors indicate real-time analysis of fluorescence lifetimes during endoscopy is one of the drivers for this work however the authors never present the time it time it takes to process an endoscopy acquired lifetime image with this method as currently presented so a reader can judge how close things are at present to "real-time".

Response 3: In the text, it mentioned that the processing time to acquire a FLT image is 0.9 seconds, and based on the surgeouns claim, a frame rate of 1 sec is sufficient for real-time.

----------------------------------------------------------------------------------------------------------------

SPECIFIC COMMENTS

Comment 1,3,4,6,7,9: 

Response : Corrected.

----------------------------------------------------------------------------------------------------------------

Comment 2: When describing the tauCAM (Lines 88-94), please provide a description of the spectral response in addition to the quantum efficiency at 780 nm.  Is it relatively flat from 700-900?

Response 2: Since tauCAM is fully explained in our latest IEEE paper and its operation is out of the scope of this paper, we refer for more details to our latest publication.

----------------------------------------------------------------------------------------------------------------

Comment 5: IRF is never explicitly defined.

Response 5: It is explicitly defined in the introduction with the capital latters.

----------------------------------------------------------------------------------------------------------------

Comment 8: How are the 6 time points used for the training data spaced? It's clear that they start at the decay peak, but are they evenly spaced all the way to the tail? 

Response 8: It is spaced evenly starting from the peak through the tail of the decay.

Reviewer 2 Report

Comments and Suggestions for Authors

Pooria Iranian and co-authors report fluorescence lifetime endoscopy with a nanosecond time-gated CAPS camera. They explored the application of this advanced imaging technology in endoscopic settings, highlighting its potential to achieve high-speed NIR fluorescence lifetime measurements. This work is interesting and suitable for this journal after some modifications.

1) Polish the English writing, especially an interesting tittle and the conclusion (section 7) for clarity.

2) Please note all the authors and the affiliations scientifically.

3) Correct the captions of the Figures and update the references styles following the Journal requirements.

4) In equation 3, the authors mentioned the IRF signals theoretically, N is set to be 4 such that it approximately matches the actual measured profiles. Since all the parameters of fluorescence decay are well understood, using synthetic data to train the CNN is a feasible approach. Whether it is necessary to correct the IRF signal following the typical TCSPC method or the IRF is a calculated constant.

Comments on the Quality of English Language

1) Polish the English writing, especially an interesting tittle and the conclusion (section 7) for clarity.

2) Please note all the authors and the affiliations scientifically.

3) Correct the captions of the Figures and update the references styles following the Journal requirements.

Author Response

Comment 1: Polish the English writing, especially an interesting tittle and the conclusion (section 7) for clarity.

Response 1: Corrected

Comment 2: Please note all the authors and the affiliations scientifically.

Response 2: Corrected

Comment 3: Correct the captions of the Figures and update the references styles following the Journal requirements.

Response 3: Corrected

Comment 4: In equation 3, the authors mentioned the IRF signals theoretically, N is set to be 4 such that it approximately matches the actual measured profiles. Since all the parameters of fluorescence decay are well understood, using synthetic data to train the CNN is a feasible approach. Whether it is necessary to correct the IRF signal following the typical TCSPC method or the IRF is a calculated constant.

Response 4: Could you please explain it more what exactly mean, because I could not fully get what you mean.